# The Roles of WNT Signaling Pathways in Skin Development and Mechanical-Stretch-Induced Skin Regeneration

**DOI:** 10.3390/biom13121702

**Published:** 2023-11-24

**Authors:** Ruoxue Bai, Yaotao Guo, Wei Liu, Yajuan Song, Zhou Yu, Xianjie Ma

**Affiliations:** 1Department of Plastic Surgery, Xijing Hospital, Fourth Military Medical University, Xi’an 710032, China; 2Department of The Cadet Team 6, School of Basic Medicine, Fourth Military Medical University, Xi’an 710032, China

**Keywords:** WNT signaling pathway, β-catenin, skin development, wound repair, mechanical stretch

## Abstract

The WNT signaling pathway plays a critical role in a variety of biological processes, including development, adult tissue homeostasis maintenance, and stem cell regulation. Variations in skin conditions can influence the expression of the WNT signaling pathway. In light of the above, a deeper understanding of the specific mechanisms of the WNT signaling pathway in different physiological and pathological states of the skin holds the potential to significantly advance clinical treatments of skin-related diseases. In this review, we present a comprehensive analysis of the molecular and cellular mechanisms of the WNT signaling pathway in skin development, wound healing, and mechanical stretching. Our review sheds new light on the crucial role of the WNT signaling pathway in the regulation of skin physiology and pathology.

## 1. Introduction

WNT signaling pathways are crucial aspects of cellular biology and have been evolutionarily conserved across different species. These pathways have a significant impact on gene expression and play a role in regulating the cytoskeleton and mitotic spindle [1]. In addition to coordinating complex cellular behavior during development, WNT signaling pathways also control cell proliferation, stem cell maintenance, cell fate decisions, organized cell movement, and tissue polarity establishment during skin wound repair and mechanical stretching [2]. In this paper, we discuss the therapeutic potential of WNT signaling in skin development, wound healing, and mechanical stretching, in addition to its molecular and cellular mechanisms in these processes.

## 2. Overview of WNT Signaling Pathways

### 2.1. WNT Signaling Pathways

Secreted WNT family signaling proteins bind to transmembrane frizzled protein (FZ) receptors on the cell membrane to form WNT/FZ complexes, which activate intracellular signaling pathways. Depending on the signaling pathways activated by different WNT/FZ complexes, they are generally classified as canonical or non-canonical WNT signaling pathways [3]. Canonical WNT signaling pathways usually consist of four key components, namely low-density lipoprotein receptor-related protein 5 or 6 (LRP5/6), FZ, WNT, and β-catenin [4] (Figure 1).

As a cell surface endocytotic receptor, LRP5/6 is indispensable in the canonical WNT signaling pathway [5]. LRP5/6 is a single-pass transmembrane protein with an extracellular domain and contains four β-propeller region/epidermal growth factor (EGF) repeats and three tandem ligand-binding repeats [6]. WNT ligands and antagonists can bind to the four β-propeller/EGF regions of LRP5/6. Then, the bound LRP5/6/WNT/FZ complex causes a conformational change in the tail of LRP5/6; this change subsequently results in phosphorylation and axon-binding of LRP5/6, ultimately activating the canonical WNT pathway [6,7]. The FZ is a cell surface receptor consisting of an extracellular WNT-binding cysteine-rich domain, a transmembrane domain of seven helices, and a cellular tail. A total of 10 FZs have been identified in *mice* and *humans*. The canonical WNT protein binds to the FZ receptor and activates β-catenin/T-cell factor (TCF), whereas the non-canonical WNT protein binds to the FZ receptor and activates the small Rho GTPase, c-jun N-terminal kinase, and other β-linker-independent signaling events [2]. The WNT protein, a secreted glycoprotein encoded by a highly conserved gene family, attaches to receptor complexes consisting of FZ family receptors and/or co-receptors to initiate the WNT signaling pathway [8]. Various WNT proteins are expressed in different spaces and times and play different roles [8]. So far, 19 distinct WNT protein-coding genes have been found in *mice* and *humans*. Among them, WNT1, WNT3, WNT3a, WNT7a, WNT7b, WNT8, and WNT10b can activate the canonical WNT signaling pathway [3,4,9,10]. The key to stable exportation of the WNT/β-catenin pathway is β-catenin. Β-catenin is both an effector of mechanical signals and a cytoplasmic/nuclear protein, and it has two forms in the plasma membrane, namely the E-calmodulin/β-catenin/α-catenin complex and free β-catenin [8,11]. When no WNT signal is present, β-catenin binds to E-calmodulin and α-catenin complexes through adherens junctions and participates in intercellular adhesion, migration, and cell–cell adhesion mechanotransduction [12]. When WNT signaling is present, β-catenin is a core member of the WNT/β-catenin signaling pathway and promotes the transcription of target genes, thereby regulating cell proliferation [11].

In the absence of WNT signaling, β-catenin in the cytoplasm forms a complex with several other proteins, including adenoma polyp protein (APC), Axin, casein kinase (CK), and glycogen synthase kinase-3β (GSK-3β). This complex leads to the phosphorylation and degradation of β-catenin through ubiquitination and proteasomal degradation [13]. WNT signaling, however, can prevent this degradation by affecting the cytoplasmic proteins that regulate β-catenin stability. The binding of WNT to the FZ/LRP5/6 complex results in the phosphorylation of Dishevelled (Dsh) and the recruitment of Axin to phosphorylated Dsh. This leads to the inhibition of GSK3 activity and prevents β-catenin degradation via the APC/Axin/GSK-3β complex [14,15,16]. B-catenin accumulation in the cytoplasm, triggered by the presence of WNT signaling, leads to the transfer of β-catenin into the nucleus through nuclear pores. Once in the nucleus, β-catenin forms a complex with TCF/lymphocyte-enhancing factor (LEF) and converts the TCF repressor complex into a transcriptional activator complex [15]. This complex activates the transcription of genes such as *c-myc* and *cyclin D1*, which promotes cell proliferation and helps maintain stem cell communities [17,18].

Non-canonical WNT signaling can be categorized as WNT/calcium (Ca^2+)^ or WNT/planar cell polarity (PCP) signaling pathways. In WNT/Ca^2+^ signaling, WNT ligand–receptor interactions lead to the release of intracellular calcium, thereby activating calmodulin-dependent protein kinase II (CaMKII), calcineurin (CaN), or protein kinase C (PKC). Among these effectors, CaMKII triggers the TAK1-NLK cascade, which suppresses the transcriptional activity of WNT/β-catenin signaling. WNT/PCP pathways involve the activation of small GTPases Rho, Rac, and Cdc42 and their downstream JNK signaling, which regulate cytoskeleton rearrangement and planar cell polarity (PCP). Non-canonical WNT pathways also play an important role in skin development and disease occurrence [8].

Studies have shown that during skin morphogenesis, the WNT/β-catenin signaling pathway determines the formation of hair placenta and dermal papillary precursor dermal agglutinates. Subsequently, in a mature individual, the WNT/β-catenin signaling pathway maintains hair regeneration in hair follicle precursor cells and dermal papilla. In addition, WNT/β-catenin regulates basal layer cell proliferation to maintain skin homeostasis in a pathological state [19]. As evidenced, the WNT/β-catenin signaling pathway has a vital function in skin development and physiological state maintenance. Hence, the following review summarizes the mechanisms of the WNT/β-catenin signaling pathway in skin development, wound repair, and mechanical stretching.

### 2.2. Regulation of the WNT Signaling Pathway

The WNT signaling pathway plays an important role in regulating skin development and maintaining homeostasis. A better understanding of the complex regulation of this pathway may have important implications for the treatment of skin-related disorders. The WNT signaling pathway is regulated by various factors including dickkopf (DKK), secretory frizzled-related proteins (SFRPs), and adenomatosis polyposis coli down-regulated 1 (APCDD1).

#### 2.2.1. DKK

DKK belongs to the secretory WNT inhibitor family and has four members (DKK1–4) [20,21]. By binding to and internalizing LRP5/6 coreceptors on the surface of cells, DKKs weaken WNT/β-catenin signal transduction [6]. DKK1 can diffuse in vivo and exhibits an extremely powerful WNT inhibitory effect [22,23]. In early development, the ectopic expression of DKK1 in the skin results in the loss of expression of β-catenin and LEF-1 in the dermis and terminates subsequent basement membrane formation [22]. In contrast, DKK1 expression is low in interfollicular skin [22]. DKK4 is a potential regulator of WNT signaling, not only during the morphogenesis of HF, but also in other ectodermal appendages [23]. The specific mechanism behind this may be that DKK4 facilitates the transition from classic WNT signals to non-classic WNT signals [24]. However, not all DKKs can suppress the classic WNT signaling pathway. DKK2 is an environment-dependent WNT inhibitor. The expression levels of different DKK receptors determine DKK2′s ability to act as both an activator and an inhibitor of the WNT/β-catenin signaling pathway [23,25]. In addition, DKK3, which has the lowest homology compared to other DKKs, does not exhibit an inhibitory effect on the WNT signaling pathway [23,26].

#### 2.2.2. SFRPs

SFRPs are glycoproteins that contain frizzled cysteine-rich structural domains, which are highly homologous to FZ receptors [27]. They are powerful signaling molecules that function upstream of WNT signaling. SFRPs have multiple biological roles in different cellular processes, including tissue development and tissue homeostasis [28,29]. Due to the presence of frizzled cysteine-rich structural domains, they can bind to FZ receptors or WNT ligands, making them effective WNT signaling regulators [30]. There are five proteins in the SFRP family: SFRP1, SFRP2, SFRP3, SFRP4, and SFRP5. These SFRPs mainly function as antagonists in similar ways (Table 1), while SFRP2, a crucial member of the SFRP family, can act as both an antagonist and an agonist of WNT signaling. SFRP2 overexpression chelates WNT ligands to prevent WNT ligands binding to FZ receptors and reduce β-catenin levels, thereby preventing WNT/β-catenin pathway overactivation and inhibiting cell proliferation and migration. However, it has also been found that SFRP2 may exert an agonistic effect on the WNT signaling pathway by directly binding to the FZ receptor [31,32]. SFRP1, SFRP3, SFRP4, and SFRP5 are all WNT signaling antagonists. Overexpression of these antagonists can inactivate the WNT/β-catenin signaling pathway. Among them, SFRP1 and SFRP5 are highly similar structurally to FZ receptors and inhibit WNT signaling by binding to WNT proteins and FZ receptors [27,33,34,35,36,37].

#### 2.2.3. *APCDD1*

*APCDD1* is a gene that is mutated in *human* hair and skin disorders. It encodes a membrane-bound glycoprotein that can be abundantly expressed in *human* HFs. The APCDD1 protein can interact with WNT3A and LRP5, two important components of WNT signaling. APCDD1 binds to LRP5 to form a complex and decreases WNT signaling outputs upon activation by WNT3a ligands. APCDD1 is also the intersection point of the WNT/BMP pathway. APCDD1 can coordinate WNT/BMP activation, which may dynamically explain periodic sequential WNT/BMP activation during the hair cycle [41,42].

## 3. The Role of WNT Signaling in Skin Development

Mammalian skin is composed of three main layers, namely the epidermis, dermis, and subcutaneous tissue. The epidermis and its derived appendages, such as hair follicles (HFs), sebaceous glands (SGs), and sweat glands (SwGs), work together to protect the body from environmental stress. The underlying dermis contains nourishing blood vessels and protein fibers. Subcutaneous tissue is adipose tissue that provides thermal insulation and energy resources [8]. WNTs play an important role in numerous cellular processes during skin development. Several molecules can regulate the WNT signaling pathway to play an essential role in the development of fibroblasts, epidermal stem cells (ESCs), and hair follicle stem cells (HFSCs).

### 3.1. Epidermal Development

In the early stage of skin development, communication between the embryonic epidermis and dermis is essential for basement membrane formation, epidermal stratification, and HF induction [8,43]. After gastrulation, embryonic cells differentiate into the epidermis and dermis. WNT signaling directs ectodermal cells to form the skin epithelium, which inhibits the ectodermal response to fibroblast growth factors (FGFs). In the absence of FGF signaling, ectodermal cells can express bone morphogenic proteins (BMPs) that guide cell differentiation into K8/K18 keratin-expressing cells, known as keratinocytes (KCs), forming the basal layer of the embryonic epidermis [44]. The basal layer of the epidermis produces a basement membrane—the physical boundary between the epithelium and the dermis—which is rich in extracellular matrix proteins and growth factors [45]. The KCs then differentiate into intermediate layer cells, which mature into heckle and granular cells before finally forming the KC envelope to execute the skin’s barrier function [46]. The development of the epidermis depends on strong WNT/β-catenin signaling. The stem cell properties of epidermal basal cells are related to WNT/β-catenin pathway activity [8]. The interfollicular epithelium (IFE) continues to grow and multiply to form new epidermis, and basal layer stem cells (SCs) continue to replenish the IFE via the hierarchical and stochastic models [47,48]. However, β-catenin overexpression in basal cells leads to serious overproliferation in the epidermis [49]. During epidermal development, WNT ligands can inhibit WNT/β-catenin signaling. WNT5a phosphorylates receptor-related orphan receptor α (RORα). The phosphorylated RORα then binds to β-catenin to form a transcription complex that inhibits WNT/β-catenin transcriptional activity [50]. In addition, WNT5a can induce ROR2–DVL interactions which negatively regulate the transcriptional activity of WNT/β-catenin signals by activating ROR2 [51].

### 3.2. Dermal Development

During development, the mesoderm divides into somites. Then, somites form the inner layer of the dermis (with proliferative potential) and an upper layer of differentiated cells [8,52]. Meanwhile, WNT signals also direct mesenchymal cells to form the dermis [52,53]. The dermis is mainly composed of dermal fibroblasts (DFs) and the extracellular matrix (ECM). DF differentiation and mutual signals with epidermal KCs are an integral part of skin formation and appendage development [54,55]. During the embryonic stage, DF precursors migrate from the somite to the subepidermis and subsequently differentiate into DF progenitors [54,56]. Then, DF progenitors differentiate into papillary fibroblast progenitors (PPs), reticular fibroblast progenitors (RPs), and hair dermal papillary fibroblasts (DPs). PPs differentiate into papillary fibroblasts and DPs, which are essential for communicating with epidermal signals and stimulating HF morphogenesis [57,58]. RPs differentiate into dermal white adipose tissue (DWAT), which helps insulate the skin, and reticular fibroblasts, which secrete dense collagen fibers that provide elasticity to the skin [57,59]. According to research by Gupta et al., DF progenitors which express WNT signaling can produce PPs and dermal condensates, which then differentiate into DPs and help to initiate skin HF development [60,61]. In addition, WNT signaling and BMP signaling are required to jointly maintain DP-induced HF formation [48]. Many key transcription factors and signaling factors are involved in DF progenitor cell development. Among them, WNT/β-catenin signaling regulates several transcription factors essential for DF development, such as Lef1, En1, Msx1, Msx2, Twist1, and Twist2 [62,63,64]. The transcription factor LEF1 expresses embryonic and neonatal papillary fibroblasts and is absent in adult fibroblasts [56]. LEF1 plays a crucial role in fibroblast development and guides different fibroblast cell lines to produce WNT/β-catenin-specific responses [64].

### 3.3. Development of Skin Appendages

#### 3.3.1. HF Formation

HF formation is a complex process resulting from interactions between the embryonic epidermis and dermis. These interactions are brought about through three stages, comprising hair placentation, hair organogenesis, and cell differentiation [65,66]. In the early stages, dermal fibroblasts receive WNT signals from the embryonic epidermis and respond by producing their own WNT signals, which in turn cause epidermal basal cells to gather and form a hair bud. The developing placenta produces more WNT ligands, which cause mesenchymal cells to form dermal condensates and the hair placenta. These cells rapidly divide and wrap around the dermal condensate, forming the HF dermal papilla [8]. Epidermal cells continue to penetrate the dermis and differentiate into mature HF inner root sheaths and hair shafts [52]. SCs in the HF can be divided into two groups: HFSCs located in the outer layer of the bulge, and stem cells located in the secondary hair germ below the bulge [47]. WNT/β-catenin signals are crucial for HFSC specification and differentiation; overexpression of these signals in embryonic epidermal cells abolishes HFSC specification and inhibits stem cell marker expression [67,68]. However, WNT5a overexpression in developing skin inhibits HFSC formation, while WNT10-mediated WNT activation increases the proportion of CD34+ HFSCs and results in enlargements of hairballs, hair shafts, and the dermal papillae [69,70].

#### 3.3.2. Development of SGs and SwGs

The development of SGs and SwGs is highly related to the development of HFs. The WNT/β-catenin signaling pathway can regulate this process. During SG development, the downstream mediators of WNT/β-catenin signaling and hedgehog (Hh) signaling are regulated by TCF3/Lef1 transcription factors, thereby affecting cell proliferation and differentiation [71]. The WNT signaling pathway regulator DKK4 exhibits high expression levels during SwG growth, which inhibits the traditional WNT signaling pathway [72]. In addition, Eda signaling and Shh signaling are also involved in the regulation of SwG formation [72]. It is clear that the WNT signaling pathway induces initial skin development and plays an important role in basement membrane formation, epidermal stratification, and HF induction. Different signaling molecules either positively or negatively regulate the WNT/β-catenin signaling pathway to maintain an appropriate expression level suitable for skin cell development. Although much has been discovered, there are still several outstanding questions. Is the development of other cells located in the skin associated with the WNT/β-catenin signaling pathway? What are the specific WNT ligands that initiate skin morphogenesis? How does the level of WNT/β-catenin signaling differ among different skin cell lineages?

## 4. The Role of WNT Signaling in Skin Wound Repair

Wound repair consists of three overlapping stages: inflammation, proliferation, and remodeling. During this process, various kinds of cells proliferate, differentiate, migrate, and die, thereby restoring the skin’s barrier and mechanical properties and rebuilding the skin. Many signaling pathways, including WNT/β-catenin, Notch, hedgehog, and various growth factor/cytokine pathways, are activated. WNT signaling is involved in epithelial construction and dermal compartment reconstruction, with β-catenin serving as an important regulatory factor [43]. The WNT/β-catenin signaling pathway mainly affects the proliferation phase.

### 4.1. WNT Signaling and Inflammatory Responses

Wound healing is initiated via bleeding and inflammation. After damage, blood vessels constrict, platelets gather, and fibrin clots develop in the wound area [43]. Platelets release cytokines that induce the migration of inflammatory cells such as neutrophils, macrophages, and lymphocytes to fibrin clots [73]. These inflammatory cells not only remove the bacteria, foreign bodies, and dead cells in the wound, but also release proinflammatory cytokines, growth factors, and vascular endothelial growth factors [74,75]. These cytokines increase vascular permeability and stimulate fibroblasts and epithelial cells to move to the wound and increase their activity, thus providing conditions for skin cell proliferation [14,76]. It has been demonstrated that during the inflammatory phase of wound healing, the release of cytokines by inflammatory cells is linked to an increase in WNT signaling, but the precise mechanism is yet unknown [14].

### 4.2. WNT Signaling in the Proliferative Stage of Wound Healing

During the proliferative phase of skin wound healing, fibroblasts, smooth muscle cells, and endothelial cells infiltrate the wound, and capillaries grow to form granulation tissue [14]. At this stage, the main manifestations are epidermal re-epithelialization and dermal reconstruction [77]. In the epidermis, KCs migrate, proliferate, and differentiate, thereby closing the epithelial space and restoring the epithelial barrier function [78]. Meanwhile, HFSCs contribute to epithelial re-formation [79]. In the dermis, fibroblasts migrate and proliferate, thereby repairing the dermis. These fibroblasts also release growth factors and secrete extracellular matrix components such as fibronectin and type III collagen, which provide mechanical strength to heal the wound [43,77]. At this stage, the local WNT response is enhanced in the wound, and WNT/β-catenin signaling can affect wound healing by regulating the cellular behavior of KCs, HFSCs, and fibroblasts [43] (Figure 2).

#### 4.2.1. WNT Signals in KCs

There are two main types of cell adhesion in differentiated KCs: desmosomes and adherens junctions. Desmosomes are multi-molecular complexes whose main components include desmocollin, desmoglein, plakoglobin, plakophilin, and the plakin family protein desmoplakin [80]. Adherens junctions are characterized by the presence of E-cadherin, α-catenin, β-catenin, and γ-catenin (plakoglobin) in the membrane [81]. When the WNT pathway is dormant, β-catenin participates in the adhesion connection. In cases of injury, active WNT signals prevent β-catenin degradation in the cytoplasm via phosphorylation and ubiquitination, and transfer β-catenin to the nucleus, which then promotes KC proliferation and differentiation through the combination of TCF/LEF complexes. Activated WNT signals can also disrupt the connections between KCs and create conditions for cell migration [82,83]. Thus, β-catenin stability inhibits KC migration and wound healing and also suppresses the repair and activation of its downstream target gene *c-myc*. In addition, β-catenin also indirectly affects wound repair and regeneration by blocking the action of other growth factors and cytokines [84]. Therefore, controlling the WNT/β-catenin pathway to promote wound healing is extremely important from a therapeutic standpoint.

#### 4.2.2. WNT Signals in HFSCs

In mature HFs, HFSCs in the telogen can be stimulated and activated by WNT signals to induce cell proliferation [85]. The WNT protein can activate HFSCs through the WNT/β-catenin pathway. It induces HFSC self-renewal and proliferation, and the differentiation of HFSCs into DP cells, hair follicle dermal sheath (DS) cells, and fibroblasts. DP cells and DS cells repopulate the DP and DS, which induces the regeneration of HFs and the skin [86]. Differentiating fibroblasts play a more important role in wound skin regeneration. Differentiating fibroblasts can not only bind to neonatal HFs to generate true DP/DS cells to support sustained HF regeneration, but they can also participate in interstitial reconstruction and promote dermal regeneration [87,88]. In addition, the WNT/β-catenin pathway can also promote the recruitment of HFSCs and fill SGs. SG cells and HFSCs filled with SGs can further differentiate into sebaceous cells [89,90]. These biological behaviors play a positive stimulating, and thus contributory, role in wound repair. In addition, wound repair is also affected by the hair cycle [91]. During the growth phase, WNT signal expression reaches its highest level and HFSCs are activated. These cells and their progeny help improve wound repair during growth. However, the level of WNT signal expression decreases to a minimum during the stationary phase [92]. Some scholars have found that the stationary and activation phases of HFSCs are regulated by the dynamic balance of BMP and WNT signaling in niche cells [47]. The expression level of WNT signals is adjusted by the dynamic BMP expression in normal skin [16]. BMP can be expressed in K6+ cells, dermal fibroblasts, and mature subcutaneous adipocytes in the telogen [58]. All of these BMPs produce inhibitory signals in HFSCs. The larger dermal environment reduces BMP expression at the end of the telogen, allowing HFSCs to receive WNT/β-catenin signals and initiate the anagen phase [8]. In addition, conditional ablation of BMP receptor-1a (BMPR-1a) also up-regulates the WNT7b promoter and down-regulates the WNT antagonist, resulting in an increase in the typical WNT signal and inducing excessive proliferation and expansion of SCs. The dynamic balance of BMP and WNT signals ensures the periodic activation of HFSCs and coordinates hair differentiation during the hair cycle [16]. In addition to WNT signals, AKT can also mediate β-catenin signal activation in HFSCs. Macrophages play an important role in this process. Injury promotes the production of the chemokine CCL2 by HF keratinocytes, which in turn recruits macrophages [93]. The number of macrophages increases during the HF growth phase and, among them, CX3CR1 bone marrow-derived macrophages secrete TNF and TGFβ1. TNF activates HFSCs through the AKT/ β-catenin signaling axis to induce wound-induced hair anagen cell re-entry/growth (WIHA) and wound-induced hair follicle neogenesis (WIHN). TGFβ1 signaling is essential for WIHA/ WIHN and may activate HF regeneration through the AKT/PI3K pathway [94,95,96,97].

#### 4.2.3. WNT Signals in Fibroblasts

The WNT/β-catenin pathway is inhibited in fibroblasts, but skin damage can activate the WNT signaling pathway [98]. It was found that during the proliferative phase of wound healing, the WNT/β-catenin pathway is activated, and the expressions of β-catenin and its target genes are up-regulated, thus increasing the proliferation capacity of fibroblasts and promoting collagen production and its more orderly and regular arrangement [99]. Elevated β-catenin levels also promote increased dermal collagen deposition and myofibroblast formation, which benefit extracellular matrix remodeling [100]. During the wound repair process, fibroblast proliferation and migration are regulated by various cytokines and signaling pathways [101]. In fibroblasts, skin injuries can activate FGF-9, which increases WNT activity in the dermis. In the early stage of wound repair, transforming growth factor-β1 (TGF-β1) activates β-catenin through the ERK pathway, leading to increased β-catenin expression and promoting wound repair and regeneration [102]. Basic fibroblast growth factor (bFGF) can reduce the effect of WNT signaling on cell proliferation, which in turn inhibits the normal growth and healing of wounds [103]. In addition, the β-catenin activity in dermal fibroblasts is regulated by fibronectin. Fibronectin activates β-catenin through a GSK3β-dependent β1 integrin-mediated pathway [77].

### 4.3. WNT Signaling in Remodeling

The final stage of wound healing is remodeling, where a collagen scar replaces granulation tissue [43]. This process includes ECM reorganization and modification, myofibroblast formation, wound contraction, and apoptosis [43]. In this stage, fibronectin and type III collagen in the ECM are degraded, type I collagen synthesis increases, and fibrous tissue bundles are formed [43,78]. Fibroblasts differentiate into myofibroblasts with contractile properties, causing wound contracture and reducing the scar’s surface area [43,104]. Together, these changes lead to wound contraction and scar formation [74]. During the remodeling phase, fibronectin in the ECM can bind SFRP4, a WNT inhibitor, to facilitate SFRP4 phagocytosis and degradation by macrophages, thereby driving chronic WNT activity [34]. The WNT/β-catenin pathway can not only induce myofibroblast differentiation, but also promote fibrogenesis, thereby repairing wounds [34,105]. However, WNT/β-catenin pathway hyperactivity will result in pathological scarring. Studies have shown that *human* hypertrophic scars and keloids exhibit high levels of β-catenin [106,107]. Akmetshina et al. found that TGF-β up-regulates canonical WNT signaling by down-regulating the WNT antagonist DKK1 [108]. In addition, R-pondin2, an agonist of WNT/β-catenin signaling, can thicken the epidermis, which is another mechanism of keloid formation [91,109]. These studies suggest the role of WNT/β-catenin signaling in wound healing.

Activation of WNT signaling can promote the multiplication and migration of KCs, fibroblasts, and HFSCs, thus repairing wounds. However, several questions remain to be resolved. 1. The above conclusions are mostly based on murine studies. Are the same effects present in *human* skin? 2. There are no conclusive studies to prove that WNT signaling can direct cell migration by affecting β-catenin in the adherens junctions. 3. In addition, TCF/LEF/β-catenin also regulates the transcription of other genes, and further analyses of other components involved in the wound repair process are needed.

## 5. Role of WNT Signaling in Mechanical-Stretch-Induced Skin Regeneration during Tissue Expansion

Skin soft tissue expansion is a surgical procedure that involves planting a silicone expander under the skin and regularly filling it with physiological saline, finally increasing the silicone expander’s skin surface area [110]. Skin soft tissue expansion creates “additional skin tissue” that is similar in texture, color, and structure to adjacent healthy skin [111,112]. Therefore, skin soft tissue expansion has been widely used to treat large skin defects such as in breast reconstructions, ear reconstructions, burn deformities, bone transplantation, and congenital giant melanocytic nevus removal [113,114,115]. However, there are still some shortcomings that limit the development of its clinical applications, such as a low expansion efficiency, thinning of the expanded skin, and local ischemia of the expanded skin [110]. To improve the speed and efficiency of skin expansion, it is important to understand the mechanisms of skin soft tissue renewal and expansion. The skin expansion process involves repeated microtrauma and reparative expansion [116]. During expander enlargement, the biological responses generated by mechanical stretching include biological growth, elastic stretching, displacement, and mechanical creep. Among them, biological growth is the most important biological reaction in the generation of new skin [113]. This is attributed to complex mechanical regulation. The force applied by the tissue expander changes the shape of the cells in the skin, which leads to alteration of local cellular molecules. The ion channels in the cell membrane and intracellular second messengers generate complex signaling pathways in response to mechanical stimuli, which convert the physical activity into a biological response [117]. Then, biological signals affect gene expression by initiating signaling cascades, thereby enhancing cell proliferation, migration, and viability. Finally, new “extra” skin tissue is formed and transferred to repair the wound [118,119,120]. Studies have shown that the WNT signaling pathway participates in mechanical transduction [121,122]. Mechanical force can promote skin regeneration by activating the WNT signaling pathway [123].

In the skin soft tissue expansion process, the epidermis, dermis, blood vessels, and skin accessory structures at the expansion site will change. In the epidermis, the cell thickness and density are increased [124]. Takei et al. found that in an in vitro pull assay, the proliferation and migration rates of *human* KCs exhibited significant increases [125]. In turn, this was related to WNT signaling activation and β-catenin accumulation in basal KCs induced by mechanical stretching [11]. Acute stretching directly stimulates β-catenin in the nucleus, which induces KC proliferation. In addition, Ledwon et al. observed that Langerhans cells (LCs) can secrete SFRP2, and SFRP2 plays a regulatory role in KC differentiation induced by WNT/β-catenin signaling [11,49]. In the dermis, the content and density of collagen are reduced. Collagen fibers are elongated, fractured, and in disorder, and the dermis becomes thinner [126]. Studies have shown that mechanical stimulation can increase fibroblast activity and promote their proliferation. As a result, more collagen is synthesized, and the dermal cell density is increased [126,127]. In addition, it was found that fibrocytes migrate faster when they are subjected to cyclic axial stretching in vitro. Furthermore, analysis of the pathways in stretched cells showed that stretching stimulated the WNT signaling pathway [128]. Therefore, the WNT signaling pathway promotes fibroblast proliferation and migration by transmitting mechanical stimuli to cells through cell–matrix interactions, cell junctions, and indirect cell communication. HFSCs in the bulge of HFs can differentiate into epidermal cells, sebaceous cells, neurons, and vascular endothelial cells, thereby promoting skin regeneration and hair growth [116]. Research has demonstrated that injecting HFSCs can lead to the generation of thicker skin with more proliferative cells and more collagen in a rat skin expansion model [116,129]. HFSC proliferation and differentiation are regulated by various signaling pathways, including the WNT, BMP/TGF, Notch, Shh, and FGF pathways [130]. Under stretching conditions, increased expression levels of WNT7b, WNT10a, and Lef1 and a widespread presence of nuclear β-catenin expression were observed in the hair matrix [131].

In the expanded skin, the blood vessel density increases [132]. This is related to the up-regulation of vascular endothelial growth factor (VEGF) via the WNT signaling pathway [133]. VEGF can recruit bone marrow mesenchymal stem cells (BMMSCs) and promote the differentiation of BMMSCs into vascular endothelial cells, which promotes angiogenesis. VEGF can also increase vascular permeability and improve blood and nutrient supply to the expanded skin, thereby promoting skin soft tissue expansion [113]. Also, LCs and macrophages also play an important role in skin regeneration. LCs are epidermal dendritic cells involved in maintaining the health and balance of the epidermis by regulating the adaptive immune responses to pathogen invasion [134,135]. Mechanical stretching stimulates the secretion of SFRP2 by LCs. The presence of SFRP2 leads to β-catenin accumulation in the nucleus of LCs, which activates the WNT signaling pathway. Finally, it promotes skin growth and restores the balance of the epidermis [11,120]. During expansion, macrophages can not only remove debris from the injured site, but also activate HFSCs and regulate their regenerative activity [136,137]. Chu et al. found that mechanical stretching can stimulate chemokine production and induce macrophage recruitment, which polarizes macrophages into the M2 subtype. M2 produces several growth factors, such as hepatocyte growth factor (HGF) and insulin-like growth factor-1 (IGF-1), which activate HFSCs and promote hair regeneration [131]. The balance between WNT/β-catenin and BMP-2 plays the most important role in this regeneration process [131]. However, more studies are needed to clarify how mechanical stretching regulates the production of growth factors and induces macrophage polarization (Figure 3).

In recent decades, studies have shown that mechanical forces can regulate the WNT/ β-catenin signaling pathway and change the cellular behavior of KCs, fibroblasts, HFSCs, LCs, macrophages, and other cells, sequentially promoting skin regeneration. However, some questions remain to be answered. What is the exact mechanism of HFSC-promoted skin growth upon mechanical stretching stimulation during tissue expansion? Do cells of different types follow different signal transduction pathways?

## 6. Conclusions

In conclusion, WNT signaling pathways, especially the WNT/β-catenin pathway, play a crucial role in skin development, repair, and mechanical stretching. They have the potential to be a promising therapeutic target for various skin disorders in the future. Further research into the specific mechanisms of these pathways and their activating and inhibiting molecules can improve the effectiveness of current skin treatments. Moreover, more studies are needed to understand the specific role of the WNT signaling pathway in response to mechanical stretching.

## Figures and Tables

**Figure 1 biomolecules-13-01702-f001:**
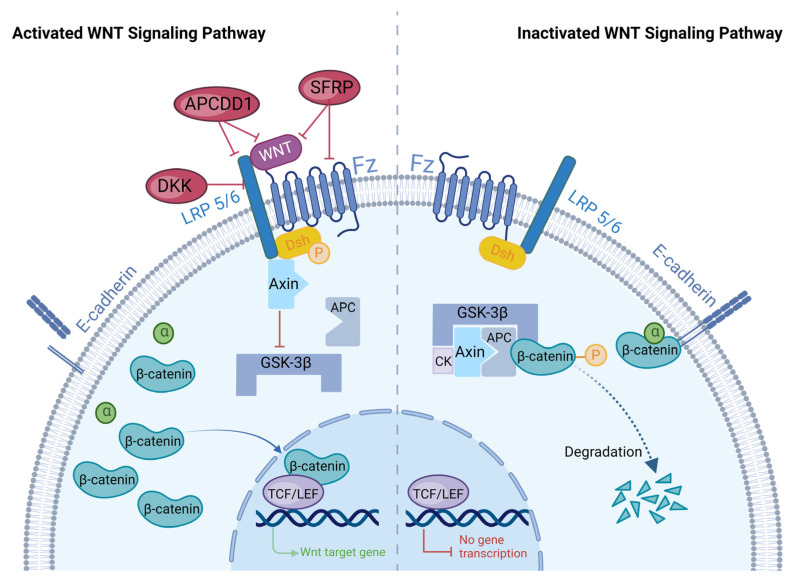
WNT/β-catenin signaling pathway. In the absence of WNT signaling, β-catenin forms complexes with several other proteins in the cytoplasm, including APC, Axin, CK, and GSK-3β, leading to the compound phosphorylation and degradation of free β-catenin in the cytoplasm. In the presence of WNT signaling, WNT can bind to the FZ/LRP5/6 complex and recruit Axin to phosphorylated Dsh. This behavior can inhibit GSK3 activity and prevent the degradation of β-catenin by the APC/Axin/GSK-3β complex, thereby increasing free β-catenin in the cytoplasm. B-catenin translocates to the nucleus through nuclear pores and forms complexes with TCF/LEF to activate downstream target gene transcription, thereby promoting cell proliferation. The primary regulators of the WNT/β-catenin signaling pathway are DKK, SFRP, and APCDD1. DKK belongs to a family of secreted WNT inhibitors that attenuate WNT/β-catenin signaling by binding to and internalizing LRP5/6 coreceptors on the cell surface. The SFRP family can combine with the FZ receptor or WNT ligands, enable antagonism or excitement, and adjust the WNT signaling pathway. APCDD1 is a membrane-bound glycoprotein in HFs. APCDD1 decreases WNT signaling by binding to both LRP5 and WNT3a.

**Figure 2 biomolecules-13-01702-f002:**
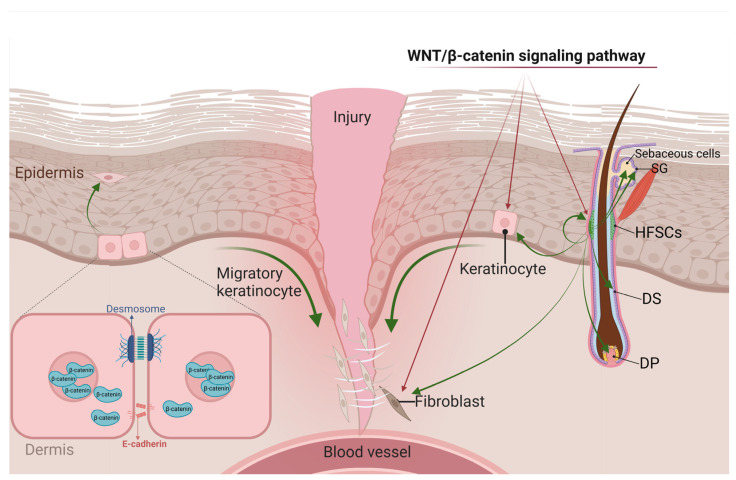
The WNT/β-catenin signaling pathway affects cell behavior in the proliferative phase of wound healing. WNT/β-catenin signaling can affect wound healing by regulating the cellular behavior of KCs, HFSCs, and fibroblasts. WNT signaling promotes KC migration by degrading E-cadherin on the surface of KCs. WNT signaling can also prevent β-catenin degradation in the cytoplasm. Based on the above, WNT signaling promotes KC proliferation and differentiation. In addition, WNT signaling induces self-renewal and proliferation of HFSCs, and WNT signaling can induce HFSCs to differentiate into epidermal cells, DP/DS cells, and fibroblasts, thereby promoting wound repair. The WNT/β-catenin pathway can also recruit HFSCs and fill SGs. SG cells and HFSCs could further differentiate into sebaceous cells. Finally, β-catenin promotes fibroblasts to produce ECM, which is required for wound closure and KC migration. β-catenin also plays a key role in fibroblast proliferation and migration.

**Figure 3 biomolecules-13-01702-f003:**
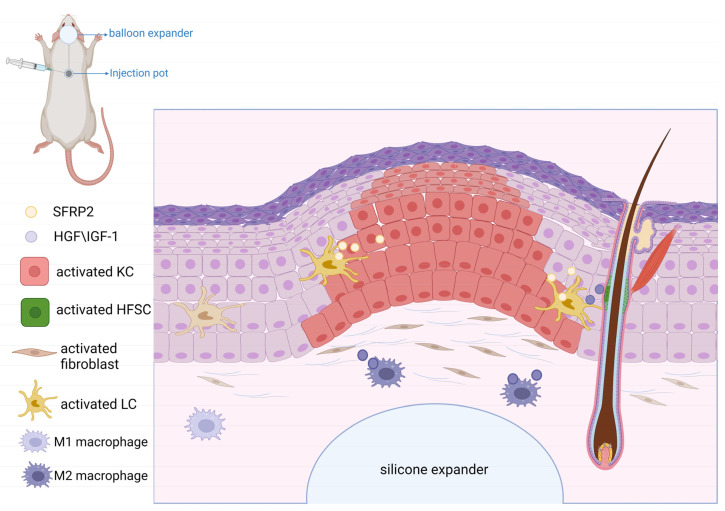
Behavior of associated cells during mechanical stretching of the skin. In the process of skin soft tissue expansion, WNT/β-catenin signaling induces changes in cell behavior at the expansion site. KC proliferation and migration increase. SFRP2 secreted by LCs induces KC differentiation. Fibroblasts increase proliferation and migration and synthesize more collagen. HFSCs differentiate into epidermal cells, SG cells, neurons, and vascular endothelial cells to promote skin regeneration. Mechanical stretching stimulates LCs to secrete SFRP2. The presence of SFRP2 leads to the β-catenin accumulation in the nucleus of LCs and activates WNT signaling. Macrophages are polarized into the M2 subtype and produce multiple growth factors that activate HFSCs and promote hair regeneration.

**Table 1 biomolecules-13-01702-t001:** Roles of DKK and SFRP in the WNT/β-catenin signaling pathway.

	Regulators	Function	References
DKK	DKK1	DKK1 inhibits WNT/β-catenin signaling by binding to LRP5/6.	[38]
	DKK2	DKK2 is an environment-dependent WNT inhibitor. The expression levels of different DKK receptors determine the ability of DKK2 to act as an activator or inhibitor of WNT/β-catenin signaling.	[25,39]
	DKK3	DKK3 does not participate in WNT/β-catenin signaling. However, DKK3 is considered to be a marker of hair follicle stem cells (HFSCs).	[21,25]
	DKK4	DKK4 transforms classic WNT signaling into non-canonical WNT signaling.	[23]
SFRP	SFRP1	SFRP1 is one of the WNT signaling pathway antagonists. The structure of SFRP1 is highly homologous to the FZ receptor and can bind WNT proteins and the FZ receptor.	[27]
	SFRP2	Overexpression of SFRP2 chelates WNT ligands and prevents the binding of WNT ligands to FZ receptors, thereby reducing β-catenin levels and preventing excessive activation of the WNT/β-catenin pathway. SFRP2 can also inhibit WNT signaling by directly binding to FZ receptors.	[31]
	SFRP3	Overexpression of SFRP3 can inactivate the WNT/β-catenin signaling pathway.	[36]
	SFRP4	SFRP4 can inhibit the WNT/β-catenin signaling pathway.	[40]
	SFRP5	SFRP5 is an inhibitor of WNT signaling. SFRP5 is structurally very similar to the FZ receptor and can inhibit WNT signaling activity by competitively inhibiting the FZ receptor.	[37]

## Data Availability

Not applicable.

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
