# Peer review of "The Roles of WNT Signaling Pathways in Skin Development and Mechanical-Stretch-Induced Skin Regeneration"

_biomolecules, 2023, doi:10.3390/biom13121702_

Round 1
Reviewer 1 Report
Comments and Suggestions for Authors
This is my Sum up and impression of the literature: The WNT signaling pathway, especially the WNT/β-catenin pathway, plays crucial roles in skin development, wound healing, and response to mechanical stretching. In skin development, WNT/β-catenin signaling directs differentiation of ectodermal cells into keratinocytes to form the epidermis and signals between the epidermis and dermis drive development of the dermis, hair follicles, and other skin appendages.In wound healing, WNT signaling promotes keratinocyte and fibroblast proliferation/migration to close wounds. It also activates hair follicle stem cells to contribute to re-epithelialization. aberrant WNT signaling can lead to pathological scarring. Mechanical stretching, such as from tissue expanders, activates WNT/β-catenin signaling which stimulates keratinocytes, fibroblasts, macrophages, and hair follicle stem cells to promote skin regeneration. WNT signaling is tightly regulated, both positively and negatively, by various modulators like DKK, SFRP, and others to maintain proper levels during skin development, homeostasis, and repair. Many details of WNT signaling mechanisms in skin remain unknown, including specific roles of WNT ligands, differences between skin cell types, and confirmation in human skin. Further research on WNT regulation of skin processes could advance treatments for skin disorders.In summary, this review covers current knowledge on the crucial and complex roles of WNT signaling in skin physiology and the need to further elucidate its regulatory mechanisms to translate findings to human skin disease therapies.
The drawing is very good, but I have many opinions on this review. I hope to follow the opinions and read the relevant literature I mentioned for correction.
1.The first problem is that Figure2 Wnt's graph is missing an extremely important connection point, DP/DS related to Wnt/β-catenin.(Rahmani et al., Development cell, 2014)( Sepideh Abbasi, Cell stem cell) (Jimin Han Front. Cell Dev. Biol., 2022).It's a very different region from Fibroblast, and there's a lot of literature about this. And DP/DS have very important relationship with Wnt/β-catenin.(Jimin Han Front. Cell Dev. Biol., 2022).
2.In the role of Wnt/ beta-catenin on the skin, two terms are very important: WIHN and WIHA. My next recommended citations will be based on these two phenomena.
The relationship between hair follicles and wound immunity has been reported in several literatures.
CX3CR1 is a Bone derived macrophage, and Ly6C is a Tissue derived macrophage. The relationship between macrophage and -TNF-α and wound needs to be explained, because their downstream pathway is β-catenin(Xusheng Wang, Nature communication,2017) (Waleed Rahmani, JID, 2018). I think it is necessary to cite several literatures to explain the relationship between Wnt/β-catenin in hair follicles and wounds(Xusheng Wang, Nature communication,2017). (Tetsuro Kobayashi Immunity,2019)
Regarding the fate between hair follicle stem cells and Wnt/ beta-catenin, I think several articles need to be added with caution(Haiyan Chen,PLOS ONE,2017)(Haiyan Chen,Theranostic,2019).
(Xusheng Wang, Nature communication,2017).
3.
About Wnt affects hair follicles to renew sebaceous glands:
This is a very new direction, to update the recent literature, we must pay more attention. Hair follicles around Wound would accelerate the renewal of sebaceous glands, and hair follicles could renew sebaceous glands in both normal hair follicles and around wound. Wnt/β-catenin is associated with new sebaceous glands(Jimin Han iscience., 2023)and Sunnywong sequenced sebaceous cells from adult mice. Happily, the article on the relationship between Wnt/ beta-catenin and hair follicles continues(Natalia A. Veniaminova, cell report,2023).
I know it's a very difficult thing for you to include all the beta-catenin articles. But updating as many new articles as possible may be very helpful.
Comments on the Quality of English LanguageMinor editing of English language required
Reviewer 2 Report
Comments and Suggestions for Authors
Bai et al tried to review Wnt signalling in skin development, wound healing and hair follicles stem cell maintenance, which might be of some values for Dermatology fields. There are rooms for further improvement to make it more clear in precise and concise ways.
Line 17, “Our findings”, not sure what it is exactly.
Line 38, “classical”, most people in Wnt field use “canonical” instead.
Line 49, “FZ is a cell surface receptor, which is formed by seven transmembrane segments and a WNT-binding site”, something like this may be more clear: FZ is a cell surface receptor, consisting of an extracellular Wnt binding cysteine rich domain, a transmembrane domain of 7 helices and a cellular tail.”
Line 57, “scholars have found “, may be simply “human has”
Line 58, Not only this three can activate canonical Wnt signalling, a lot of others, like Wnt8 etc.
Line 65-67, the reference only said SFRP2 is potentially a mechanotransduction effector, here, are you saying: When no WNT signal is present, β-catenin participates in mechanotransduction?
Line 74, “Dishevelled FZ”, do you mean Dishevelled to FZ? Whole sentence, “results in the activation of the classical WNT signaling pathway and recruitment….”, should be recruitment… lead to Wnt activation, not activation and recruitment.
Line 118 “Due to the presence of frizzled cysteine-rich structural domains, they can bind to FZ receptors or WNT ligands” why presence of CRD can bind to FZ receptors? Pleaser provide original references.
Figure 1, in Right half, why without Wnt, FZ and LRP5/6 also linked by Dsh? Is non-canonical Signaling involved in skin physiology/pathology at all? Some receptors, like APCDD1 is more linked to follicle physiology, should be mentioned.
Line 206, 224, should give a meaning full sentence, not just HF, etc. So do others, like KC.
Line 388, “WNT signaling pathway is a classical mechanical transduction pathway”, please cite original paper, not review, as there is no acceptable conclusion that “WNT signaling pathway is a classical mechanical transduction pathway”, as a review, should not make “novel” conclusion based on nothing.
Comments on the Quality of English LanguageExtensive editing of English language required
Round 2
Reviewer 1 Report
Comments and Suggestions for Authors
With my suggestion and your correction, everything is perfect.
Reviewer 2 Report
Comments and Suggestions for Authors
The revised manuscript is improved. No more questions.